# Distribution of ESBL/AmpC-*Escherichia coli* on a Dairy Farm

**DOI:** 10.3390/antibiotics11070940

**Published:** 2022-07-13

**Authors:** Timo Homeier-Bachmann, Jette F. Kleist, Anne K. Schütz, Lisa Bachmann

**Affiliations:** 1Friedrich-Loeffler-Institut, Federal Research Institute for Animal Health, Institute of Epidemiology, 17493 Greifswald-Insel Riems, Germany; al18202@hs-nb.de (J.F.K.); anne.schuetz@fli.de (A.K.S.); 2Faculty of Agriculture and Food Science, University of Applied Science Neubrandenburg, 17033 Neubrandenburg, Germany; bachmann@hs-nb.de

**Keywords:** ESBL-*E. coli*, calves, cattle, antimicrobial resistance

## Abstract

The aim of the study was to determine the prevalence of ESBL/AmpC-producing *Escherichia* (*E*.) *coli* and to investigate their on-farm distribution on an exemplary dairy farm. For this purpose, sample sizes were calculated, and fecal samples were collected from cattle of all ages and analyzed for the presence of ESBL/AmpC-*E. coli* using selective media supplemented with cefotaxime. These antibiotic-resistant bacteria were detected in 22.5% of the samples tested. The prevalence was highest in the calf age group, in which 100% of the collected fecal samples were positive. With increasing age, the prevalence decreased in the other sample groups. While ESBL/AmpC *E. coli* could still be detected in young stock (15%) and breeding heifers (5%), no resistant pathogens could be detected in adult animals. Whole-genome sequencing of the ESBL/AmpC-*E. coli* isolates revealed, first, that all isolates were ESBL producers (CTX-M-1 and CTX-M-15) and, second, that ST362, which is known as a biofilm producer, was dominant in the calves (85%, n = 17). Based on these results and the evaluation of a questionnaire, possible causes for the occurrence of ESBL/AmpC-*E. coli* were discussed and recommendations for the reduction in transmission were formulated. Unlike most German dairy farms, no waste milk feeding was apparent; therefore, factors reducing ESBL/AmpC-*E. coli* are primarily related to an improvement in hygiene management to prevent biofilms, e.g., in nipple buckets, but also to question the use of antibiotics, e.g., in the treatment of diarrheic calves.

## 1. Introduction

With the introduction of antibiotic therapies and increased contact between bacteria and antibiotic substances, the development and spread of antibiotic resistance accelerated rapidly [1,2]. Antibiotic resistance is continuously increasing, while at the same time, the number of effective antibiotics continues to decrease. The WHO now assesses the current resistance situation as a threat to global health. If the resistance situation continues to develop at such a rapid pace, this may mean that there will be no more effective antibiotics in the future [3]. Livestock farming, in particular, has been repeatedly criticized for using antibiotics too frequently and in an untargeted manner, which greatly accelerates the development of resistance [2,4]. Other factors that influence the occurrence of antibiotic resistance include farm management, water treatment, manure handling, and wildlife control [5].

Extended spectrum beta lactamase (ESBL)-producing *E. coli* pose a significant threat to global health, which is why the WHO has placed them in the critical category of multidrug-resistant pathogens, for which the development of new antibiotics is of enormous importance [6]. The resistance situation of *E. coli* is developing rapidly. In 2016, 58.6% of human-derived *E. coli* isolates from the European Union collected during 2013 and 2016 in the frame of the EARS-Net already showed resistance to at least one antibiotic agent [4,7]. ESBL-*E. coli* also play a role in dairy farming, both as a public health threat and as a potential source of environmental contamination [8]. In 2013, Schmid et al. investigated the prevalence of ESBL and AmpC-producing *E. coli* in mixed dairy and beef cattle farms and pure fattening farms in Bavaria, Germany. ESBL/AmpC-*E. coli* were detected in 32.8% of the samples collected. In total, on 39 of 45 participating farms, ESBL/AmpC-producing *E. coli* were present, corresponding to 86.7% of the farms tested. It was also found that the probability of the presence of ESBL/AmpC-producing *E. coli* was higher in mixed farms than in fattening-only farms [9].

A systematic review examined the distribution of multidrug-resistant *E. coli* in dairy cattle. In summary, several studies found that the highest detection rates of multidrug-resistant *E. coli* occurred in the age group of preweaned calves. Detections of resistant *E. coli* decrease with increasing age [10]. A study in Bavaria, Germany showed the same result for dairy farms. In addition, it was shown that for cattle-fattening farms, there was also a decrease in ESBL-producing Enterobacteria with the age of the fattening animals. However, fattening farms were overall less affected than dairy farms (18.9% vs. 39.6%) [9]. Slightly more positive results were obtained in another study, also from Germany. For dairy farms, the authors determined a prevalence of cefotaxime-resistant *E. coli* of 48% and for fattening farms of 35% [11]. Several studies comparatively investigated the prevalence of ESBL-producing *E. coli* in dairy herds. Massé et al. observed an average prevalence of 63% in calves and 19% in cows in Canadian dairy herds [12]. For Germany, Weber et al. recently reported an ESBL prevalence of 63.5% for calves and 18.0% for cows in a study conducted in a large number of dairy herds [13]. A deeper insight into the distribution of ESBL-*E. coli* within a dairy farm located in the United Kingdom is provided by Watson et al. This study was already carried out in 2012 and covers only one farm. Different groups of cattle were tested for ESBL-*E. coli*. Low levels of ESBL-positive samples were found for bulling heifers and dry cows (2.6% and 3.4%, respectively). Significantly higher levels of approximately 30% ESBL-positive samples were found in lactating cows [14]. These values from the UK for lactating cows also exceed those observed in Germany by Weber et al.

A recent study on the prevalence of sequence types (ST) of *E. coli* in cattle in the United States showed that the most common ST complexes (STC) were STC10, 58, 88, and 29. For the CTX-M-positive *E. coli*, the same STCs were predominantly found with the exception of STC29. In addition to these clustered STC, a wide range of individual detections of ST were reported (34.9% of STs present in CTX-M-positive *E. coli* were not affiliated to an STC, for CTX-M-negative *E. coli* this was the case in 48.4%). Consequently, the diversity of CTX-M-positive *E. coli* is considered to be low [15]. To our knowledge, there are no more recent studies on the distribution of ESBL-producing *E. coli* in dairy farms.

Therefore, the aim of the study was to determine the prevalence of ESBL/AmpC-producing *E. coli* and to investigate their on-farm distribution on an example farm representative for the average size of dairy farms in Eastern Germany. By identifying the sources of infection, targeted intervention measures should be derived.

## 2. Results

### 2.1. ESBL/AmpC-E. coli Isolation and Characterization

In total, 120 fecal samples were taken for this study. Samples originated from calves, young stock, breeding heifers (inseminated or pregnant), and from high- and late-lactating as well as from dry cows of one representative farm. Of these, we isolated phenotypic cefotaxime-resistant putative *E. coli* in 27 samples in the bacteriological examination. Thus, ESBL/AmpC-producing *E. coli* was suspected in 22.5% of the samples tested.

Of the 20 samples obtained from calves, 100% displayed phenotypic cefotaxime resistance. The prevalence was significantly higher than in all other age groups *(p <* 0.0001, chi-square test of independence). In young stock, 20 samples were collected and tested. Only three of these samples revealed cefotaxime-resistant *E. coli* during bacteriological examination. In total, 15% of the young cattle samples were positive. The prevalence was also significantly higher than in all age groups other than calves *(p <* 0.0001, chi-square test of independence). Likewise, 20 samples were taken from the breeding heifers of the rearing farm. Of these samples, only one sample contained a phenotypically resistant isolate, i.e., 5% of the heifers.

Fifteen samples each were taken from cows in high lactation and late lactation, and 20 samples from dry cows. None of these samples exhibited phenotypic resistance to cefotaxime.

The farm consisted of two farm sites, so that the young stock was kept separately from the other animals. Five slurry samples were collected from each of the two farm sites. One sample from the main farm site and two samples from the rearing farm site harbored phenotypic cefotaxime-resistant *E. coli.*

### 2.2. Results of the Questionnaire

The study herd consisted of 1800 animals which were kept on two locations. All heifer calves were reared; no additional purchasing of animals was necessary. Concerning known risk factors for the occurrence of ESBL/AmpC-*E. coli,* the following issues were remarkable: 1. no waste milk feeding for the calves; 2. daily cleaning of the nipple buckets with water without detergent or disinfectant; 3. antibiotic treatment of diarrheic calves with amoxicillin/clavulanic acid or enrofloxacin without microbial testing and antimicrobial susceptibility testing (Appendix A).

### 2.3. Whole-Genome Sequencing (WGS) and Analysis

In order to carry out genotypic profiling, the putative ESBL/AmpC-producing *E. coli* obtained during the study were whole-genome sequenced. Based on sequencing, ESBL genes (CTX-M type) were detected in 26 of the 27 isolates, and one isolate sample carried only AmpC genes. MLS typing detected six different sequence types.

#### 2.3.1. Multilocus Sequence Typing (MLST)

Six different STs were detected by MLST. These were the STs 362, 117, 88, 967, and two new sequence types. ST362 occurred 16 times and was isolated exclusively in calves. ST117 was detected in two calf samples and in the main farm slurry sample. The three isolates from the young cattle belonged to ST967. The detected AmpC-transmitting isolate belonged to ST88. Two new STs were detected in the MLST. One of these new STs was detected in the two samples from the rearing farm (heifer sample and one slurry sample). It showed only minimal differences from the ST58 of STC155. The second new ST belonged to the isolate from the other positive slurry sample from the rearing farm and displayed great similarity to ST10 of STC10.

#### 2.3.2. Plasmids

Colicinogenic plasmids were present in all isolates from calves. All isolates carried Col156-type plasmids and, with one exception, ColRNAI-type plasmids (isolate no. 1640). In addition, two further colicinogenic plasmids occurred: Col(pHAD28) was present in six isolates (all ST117 isolates and isolate no. 1628), and Col(MG828) occurred in isolate no. 1628. Furthermore, colicinogenic plasmids (Col(pHAD28) were also found in the isolate of the heifer from the rearing herd (isolate no. 1629). IncB/O/K/Z-type plasmids were present in all but one calf isolates (isolate no. 1628 (ST88)) and in one isolate from the slurry of the main farm (isolate no. 1646). Additionally, 19 of the 20 calf isolates carried an IncFIB-type plasmid. The exception was again the calf isolate no. 1628 (ST88). This plasmid type was further detectable in all isolates of the young cattle as well as in two slurry samples (isolates no. 1646 and 1694). Other plasmids of the IncF type occurred sporadically (IncFIA, IncFIC, and IncFII). IncFII-type plasmids were exclusively harbored by ST117 isolates, regardless of the origin of the isolates (calf feces and slurry). IncH1- and IncR-type plasmids were found only once in each case (isolates nos. 1637 and 1649). For one isolate, no plasmids at all were detectable by WGS (isolate no. 1648). Nevertheless, CTX-M-1 was detected in this isolate and a BLAST analysis of the contig containing the CTX-M-1 (length 3912 nt) revealed the highest identities with IncR-type plasmids.

#### 2.3.3. ESBL

DNA-sequencing analysis revealed that all isolates harbored AmpC1 and 2 as well as *ampH*. Additionally, we also detected two different CTX m types (CTX-M-1 and CTX-M-15) in 26 isolates. All 26 samples in which the presence of ESBL-*E. coli* could be detected contained ESBL of the CTX-M type. Eighteen of these samples contained ESBL of the CTX-M-1 type and in the remaining eight samples, ESBL of the CTX-M-15 type could be detected. Of 20 isolates from calves, 17 produced ESBL of type CTX-M-1 and three produced ESBL of type CTX-M-15. We were able to isolate CTX-M-15 in the ESBL *E. coli* from the young cattle, whereas the isolate from the heifer contained ESBL of type CTX-M-1. The isolate from the slurry from the main farm contained ESBL type CTX-M-15 and the two isolates from the slurry samples from the rearing farm contained either CTX-M-1 or CTX-M-15. In one calf, we were able to obtain a cefotaxime-resistant isolate in which only AmpC could be detected (isolate no. 1628). In addition, the ST362 isolates from the calves were all carriers of TEM-105.

#### 2.3.4. AMRG

AMRGs *floR*, *strA*, and *strB* were detected in all calf isolates belonging to STs 362 (n = 16) and 88 (n = 1), as well as in one young cattle (isolate no. 1645). A similar distribution was found for *catA*, with the difference that isolate 1645 (young cattle) did not harbor this AMRG. The florfenicol resistance gene, *floR,* was present in isolates from the heifer (isolate no. 1629) and in one slurry sample (isolate no. 1648). The isolates nos. 1648 (originating from slurry from the rearing farm) and 1629 (originating from the heifer from the rearing farm) were the only ones that possessed *aadA* and *dfrA.* Several tetracycline resistance genes were detectable in WGS analysis. The most common were *tetA*, *tetR*, and *tetY*, which have a largely consistent distribution pattern. In particular, isolates of ST362 were carriers of these AMRGs. The four ST117 isolates (three originating from calves and one from slurry) were negative for these AMRGs but positive for *tetB*. This AMRG also occurred in one ST362 isolate originating from calf feces (isolate no. 1637). The *mph(A)* gene occurred in all isolates belonging to ST362 and in the two ST58-like isolates originating from the rearing farm. Further details can be found in Table 1.

### 2.4. Antimicrobial Susceptibility Testing (AST)

Phenotypic AST was carried out using a VITEK2 apparatus (bioMérieux, Nürtingen, Germany). All 27 isolates were resistant to beta-lactam antibiotics (amoxicillin/clavulanic acid, ampicillin, cefotaxime, ceftazidime, and cefepime). The 4 isolates belonging to ST117 were resistant to ciprofloxacin and 20 isolates were resistant to trimethoprim/sulfamethoxazole. All isolates were sensitive to imipenem, meropenem, colistin, gentamicin, fosfomycin, tigecycline, and amikacin. Thus, 23 of the 27 isolates phenotypically fulfilled the definition as multidrug resistance (MDR, resistance to at least three antimicrobial classes). Details are given in Table 2.

## 3. Discussion

### 3.1. Bacteriological Examination

A total of 27 phenotypically cefotaxime-resistant *E. coli* were isolated from 120 fecal samples by bacteriological examination. Further investigations confirmed these isolates as ESBL/AmpC-producing *E. coli*. In the calf age group, 100% of the samples tested harbored ESBL/AmpC-producing *E. coli*, which represented a significantly higher proportion of resistant pathogens than in all other age groups examined. As confirmed by the chi-square test, there was a significant association between age (calves or not) and the harboring of ESBL/AmpC-producing *E. coli*. In the young-cattle age group, only 3 out of 20 samples, i.e., 15%, contained ESBL-*E. coli.* There was also a significant relationship (p-value < 0.05) between the age group of young cattle or older age group and the presence of ESBL-*E. coli*. In the heifer age group, the proportion of positive samples, with 1 out of 20, was only 5%. In the adult animals, ESBL/AmpC isolates could not be detected in either the dry cows or the high- or late-lactating cows. A decrease in ESBL/AmpC load with animal age has been reported several times [12,13,14]. Our results also show a greater colonization of suckling calves than was seen in older young animals and adult cows.

The routes of transmission of multidrug-resistant *E. coli* within and between cattle herds are not well understood. The animal age, herd size, antibiotic use, purchase of animals from other farms, the feeding of waste milk, or the hygiene of the nipple buckets for suckling calves are factors that may be responsible for the high prevalence and the spread of ESBL/AmpC-*E. coli* among calves [9,16,17,18].

In a recent study, an ESBL/AmpC-*E. coli* prevalence of more than 60% was found in calves and a prevalence of 18% in dams on large dairy farms in Germany. Risk factors identified in this study were the feeding of waste milk and the cleaning of the nipple buckets [13]. Our study shows that high burdens of ESBL-producing *E. coli* are possible in spite of not feeding bulk milk. This underlines the multifactorial nature of the AMR problem. Other studies have also demonstrated associations between cleaning and hygiene measures and the prevalence of multidrug-resistant pathogens [19,20,21].

In the present study, daily cleaning of the drinking buckets with water was performed. However, since neither detergents nor disinfectants were used, the nipple buckets should still be regarded as a risk factor, so the cleaning regime should be adapted. Unfortunately, we did not include the nipple buckets in our sampling. However, bacteriological monitoring would help to optimize the cleaning regime. In particular, prior to the use of a bucket on a new calf, intensive cleaning and, if necessary, disinfection is required to minimize the risk of spreading ESBL-*E. coli* to other calves [13]. Associations between improving cleaning and disinfection measures of feeding equipment and decreasing ESBL/AmpC prevalence have been reported in the literature [17,22].

At the farm studied, we revealed a high incidence of calf’s diarrhea treated with antibiotics unless knowing the causing pathogen, which is unfortunately a common practice in dairy herds: according to Olson et al., the probability of diarrheic calves receiving an antimicrobial treatment is higher than being administered with an electrolyte solution [23]. The frequent use of antibiotics is known to accelerate the development of antibiotic resistance. Before antibiotic treatment, on-farm testing of the major pathogens causing diarrhea (*rota* and *corona virus*, *E. coli* K99, and *Cryptosporidium parvum*) is possible [24]. Therefore, we recommend testing which is the causing agent of diarrhea. According to studies in Europe [25], Australia [26], and South America [27], *rota virus* and *Cryptosporidium parvum* are particularly responsible for diarrheic disease in neonatal calves. These pathogens are not susceptible to antibiotic treatment.

### 3.2. Whole-Genome Sequencing

Since no ESBL/AmpC-*E. coli* strains were isolated from adult animals, it is unlikely that the colonization of the calves was due to mother-to-calf transmission. The youngest calf sampled was four days old and already ESBL-positive. This suggests that calves come into contact with the reservoir for ESBL/AmpC-*E. coli* in the first days of life. Supporting this assumption, WGS revealed that IncB/O/K/Z as well as Col plasmids RNAI and 156 were exclusively present in the isolates of the calves, with the exception of the isolate from the slurry. IncB/O/K/Z-type plasmids were frequently found in *E. coli* from animal sources and are mostly associated with CTX-M-type resistance genes [28]. This is also true for the isolates investigated in this study. Additionally, a dominant ST was detected in the calves. This was the sequence type ST362; 16 of the 20 calf-originating ESBL-*E. coli* isolates belonged to this ST. The exclusive occurrence in calves indicates a very good adaptation to the calf’s environment. This might explain why, despite its high occurrence in calves, it has not been able to establish itself in older animals and their environment. Phylogroup D ST362 is classified as extraintestinal pathogenic *E. coli* (ExPEC) according to a study by Vieille et al. and is associated with various extraintestinal infections [29,30]. The dominance in calves may be due to good biofilm-sforming ability as well as increased resistance to biocides. A study on ExPEC in chickens showed that ST362 are good biofilm formers with strong swimming and swarming activity [31]. However, comparable studies in cattle are currently not available.

Every calf on the farm is fed from nipple buckets during the first month of life. The nipple buckets are often contaminated with feces, e.g., by falling down. It is possible that biofilms, particularly of ST362 ESBL-*E. coli*, may form on the surfaces of the buckets and nipples. These biofilms could then serve as a reservoir and source of colonization for the calves. The analysis of the WGS datasets revealed the presence of genes involved in the formation and development of biofilms (*csgD*, *hha*, *bcsA*, *pgaC,* and *fimB* [32]). The evaluation of the questionnaire revealed that the calves’ nipple buckets are cleaned with water once a day. However, detergent and disinfectant are not used. Weber et al. suggested that cleaning may result in redistribution of the AMR pathogens rather than elimination of them as desired. Thus, the buckets could continue to be contaminated with ESBL-*E. coli* after cleaning and additional calves could come into contact with them [13]. Heinemann et al. found a high incidence of ESBL/AmpC isolates in drinking accessories on ESBL-positive farms, especially on the inner surface of the nipples of nipple buckets, which contributed to the colonization of calves with ESBL-producing pathogens [22]. Further phenotypic studies on the biofilm formation capacity and biocide resistance of the ST362 isolates in this study are needed for clarification.

All calves are housed in calf igloos during their first month of life. These housing systems have already been demonstrated to contribute to the spread of ESBL/AmpC-producing *Enterobacteriaceae* [22]. Fecal contamination of the walls of the igloos may be inadequately removed by standard cleaning procedures, so the igloos could also serve as a reservoir. Furthermore, biocide resistance of ST362 *E. coli* could contribute to disinfection measures being limited in their effectiveness (corresponding genes were present in the WGS analysis of the isolates of the calves, e.g., *emrE*).

Three other calf isolates belonged to ST117. This ST can be frequently detected in poultry and also in humans, but less frequently in cattle, in the context of extraintestinal infections [33]. The ST117-*E. coli* detected in calves in our study exhibit some ExPEC specific virulence genes. In addition to the calves, this ST was also detected in the slurry from the main farm. This is surprising since the calves are kept in igloos on straw and thus do not produce slurry. However, the calf igloos are located adjacent to the slurry ponds; thus, an accidental introduction of these bacteria into the slurry by, for example, cleaning water, could be possible. Another option might be the presence of this ST in adult animals as well, and that it is introduced into the slurry via their feces. It is possible that this ST occurs only in low prevalence in the adult animals (far below the ESBL/AmpC *E. coli* prevalence of 18% described by [13]), so that a detection is not possible.

In addition, an ESBL-negative, but AmpC-positive, *E. coli* was detected in one calf. However, this isolate belonging to ST88 seems not to be well adapted to the environment in the calf area since no further isolates were detected in other calves or older animals.

In young cattle, we detected ESBL-*E. coli* at a low frequency. In total, three ESBL-producing *E. coli* could be isolated from the 20 fecal samples. Interestingly, this was the first time that a new ST (ST967) appeared that did not match any of the STs previously detected in the calves. Since this ST exclusively occurred in this age group, there might be some adaptation to animals of this age group. The young cattle are kept on straw until they are transitioned to slatted floors at the first insemination age. It is conceivable that the ST967 is adapted to an environment with straw and cannot maintain itself in barns with slatted floors.

The isolate from the heifer sample (isolate no. 1629) could not be assigned to any sequence type by MLST. However, a detailed analysis showed high identity with the ST58. The identical new ST also occurred in one of the two slurry samples from the rearing farm (isolate no. 1648). MLST analysis at least allows assignment to the STC155. Despite recent reports of this ST in healthy cattle at the slaughterhouse [34,35], there are nevertheless reports of involvement of this ST in clinical processes in humans and animals (bloodstream infections [36]) and cattle (mastitis [37,38]). According to a recent publication, this ST is assigned to the group of extraintestinal pathogenic *E. coli* (ExPEC) and is described as novel and globally spreading [39]. Additionally, ST58 is thought to occur in multidrug-resistant variants [36]. This ST has already been detected in cattle in Germany. The corresponding isolate also showed high agreement in WGS with the ST58-like isolates of this study; thus, *sul2* and *dfrA* as well as *mph(A)* were present in all isolates in addition to blaCTX-M1 [40]. For the slurry-originating isolate no. 1648, we could not detect any evidence of plasmids in the WGS analysis. This could indicate the chromosomal integration of blaCTX-M-1. Such chromosomal localizations of CTX-M genes have been reported several times for *Enterobactereales* (including [41,42]). Shawa et al. suggest that such chromosomal integrations in combination with other antibiotic resistance genes may contribute to the successful dissemination of the corresponding isolates [43]. However, BLAST analysis of the contig containing CTX-M-1 (length 3912 nt) revealed the highest identities with IncR-like plasmids, making chromosomal integration rather unlikely. Further investigations are needed to verify whether the blaCTX-M-1 gene of our isolate is episomally or chromosomally localized.

We suspect that the detection of the STC155 isolate in the heifer was not an isolated finding, as it is unlikely that the pathogens from a single animal were detectable in the slurry. For comparison, ST117, which was also present in slurry, was found several times in the calves. We assume that the prevalence of this ST is clearly below our detection threshold. The same is probably true for the second ESBL-producing isolate (STC10) found exclusively in slurry from the rearing farm.

Due to the few detections of ESBL-positive *E. coli* in the young cattle and heifers of the rearing site, it is hardly possible to narrow down the origin of the bacteria. The young cattle are kept on straw and all have isolates from the same ST, which, however, is no longer present in older animals. The straw originates from agricultural sites that are fertilized with chicken and pig slurry, and thus could potentially be contaminated with resistant pathogens. It is possible that ESBL *E. coli* is being introduced onto the farm through the straw. However, there are no reports in the literature of the presence of ST967 on pig or chicken farms. An investigation in the respective pig and chicken farms could provide clarity.

In contrast, the heifers are not kept on straw and, consequently, the ST967 of the young cattle is not present. The staple feed of the heifers, however, comes from the same areas as the straw, so contamination may have occurred in this feed as well. It remains unclear why different STs were detected in each case. This could be a coincidence, but it could also be the result of different times for the collection of forages and straw as well as the application of slurry as fertilizer.

STs 117 and 962 and the one ST10-like isolate from the slurry were the only isolates that were CTX-M15-positive. This CTX-M variant is predominantly associated with humans [44]. This finding agrees well with ST117, which is frequently detected in urinary tract infections in humans [45].

### 3.3. Antimicrobial Susceptibility Testing

Phenotypic resistance patterns to trimethoprim/sulfamethoxazole are congruent with WGS results. For the few phenotypic resistances to ciprofloxacin (isolates nos. 1624, 1627, 1643, and 1646; all belonging to ST117), no equivalents were found in the WGS (e.g., *qnr*). However, there are numerous other determinants of quinolone resistance that may not be included in the databases used [46]. More in-depth studies could help clarify the exact causes of quinolone resistance in these isolates, e.g., resistance due to efflux pump expression (disk diffusion method [46]). Nevertheless, this finding underlines the value of phenotypic-resistance testing.

All isolates were susceptible to the carbapenems meropenem and imipenem. These findings are not surprising, since carbapenems are not approved for use in livestock in the European Union (EU) and similar results have been reported by others (reviewed in [47]). Moreover, we did not detect any resistance to colistin which is a last-resort antibiotic for humans. This result was unexpected in that colistin has also been used in veterinary medicine in the EU in recent decades, leading to selection pressure and the development of colistin-resistant bacteria [48]. Colistin is predominantly used in pig and cattle production to control enteric infections caused by *E. coli* and *Salmonella* or for metaphylactic treatment [49,50].

## 4. Materials and Methods

### 4.1. Sampling and Transportation

Fecal samples taken for this study originated from calves, young stock, breeding heifers (inseminated or pregnant), and from high- and late-lactating as well as from dry cows. The sample size was calculated using the sample size calculation tool (https://epitools.ausvet.com.au/oneproportion, accessed on 20 June 2022), and based on the number of appropriately aged animals present on the day of sampling. According to our previous study [13], we chose to use an expected ESBL-*E. coli* prevalence of 60% for the age group of calves, and 20% for the other animals, a confidence interval of 90%, and a desired precision of 10% [51]. In cows, we focused on lactating animals (high and late lactating) during sampling. Additionally, samples from slurry were collected. All samples were taken on 1st of August 2021. The sample numbers are given in Table 3. Fecal and slurry samples were taken using swabs with amies transport medium (Sigma Transwap, MWE, Corsham, Wilts, UK). Swabs were stored at 5 °C until further use.

### 4.2. Questionnaire and Data Collection

Together with the herd manager, a questionnaire was completed. The questionnaire collected information regarding potential risk or protective factors for the prevalence, or transmission and distribution, of ESBL/AmpC-*E. coli* within the farm. The complete questionnaire can be found elsewhere [13].

### 4.3. ESBL/AmpC-E. coli Isolation and Characterization

Fecal swaps were cultured on CHROM ID agar plates (Mast Group, Reinfeld, Germany) supplemented with 2 µg/ mLcefotaxime (Alfa Aesar by Thermo Fisher Scientific, Kandel, Germany) in order to promote growth and identification of ESBL/AmpC-producing *E. coli* with high specificity. According to the manufacturer’s protocol, pink-violet-colored, shiny colonies represent ESBL/AmpC-*E. coli*-positive results. Positive colonies were then subcultivated on new CHROM ID agar plates supplemented with Cefotaxim until a pure culture was achieved. The isolates were then stored at −80 °C in 15% glycerol until further use. The chi-square test of independence was performed with python 2.7.5 using scipy.stats.chi2_contigency.

### 4.4. Whole-Genome Sequencing (WGS) and Analysis

WGS was performed for the overall 27 ESBL/AmpC suspect *E. coli* isolates. DNA extraction was performed using the MasterPure™ DNA Purification Kit for Blood, Version II (Lucigen, Middleton, Charleston, SC, USA) and subsequently quantified using QuBit fluorometer (Thermofisher Scientific, Waltham, MA, USA). DNA samples (concentration of the purified DNA at least 10 ng/µL) were then shipped to the Microbial Genome Sequencing Center (MiGS, Pittsburgh, PA, USA). Sample libraries were prepared using the Illumina DNA Prep kit and IDT 10 bp UDI indices, and sequenced on an Illumina NextSeq 2000, producing 2 × 151 bp reads. Demultiplexing, quality control, and adapter trimming were performed with BCL Convert v3.9.3 (Illumina, Inc., San Diego, CA, USA); https://support-docs.illumina.com/SW/BCL_Convert/Content/SW/FrontPages/BCL_Convert.htm, accessed on 20 June 2022).

The sequence analysis is described elsewhere [52,53]. In brief: We used BBDuk from BBTools v. 38.89 (http://sourceforge.net/projects/bbmap/, accessed on 1 June 2022) for (i) adapter trimming, (ii) filtering for contaminants, and (iii) quality trimming. Quality controlling of all reads was performed using FastQC v. 0.11.9 (http://www.bioinformatics.babraham.ac.uk/projects/fastqc/, accessed on 1 June 2022). For de novo genome assembly, we used the Shovill v. 1.1.0 assembly pipeline (https://github.com/tseemann/shovill, accessed on 1 June 2022) in combination with SPAdes v. 3.15.0 [54]. In addition to the polishing step of the Shovill pipeline, the assemblies were subjected to another polishing step using bwa v. 0.7.17 [55]. The SAM/BAM files obtained were then sorted and duplicates identified using SAMtools v. 1.11 [56]. Pilon v. 1.23 performed the variant calling [57] and CheckM v. 1.1.3 [58] was additionally used to estimate genome completeness and freedom from contamination. Thereafter, assemblies were analyzed for MLST determination and antibiotic-resistance/virulence-gene detection using the tools mlst v. 2.19.0 (https://github.com/tseemann/mlst, accessed on 1 June 2022) and ABRicate v. 1.0.0 (https://github.com/tseemann/abricate, accessed on 1 June 2022), respectively. Third-party databases (i.e., PubMLST [59], VFDB [60], ResFinder [61], PlasmidFinder [62], BacMet [63], ARG-ANNOT [64], and Ecoli_VF (https://github.com/phac-nml/ecoli_vf, accessed on 1 June 2022)) were used for the analyses of both tools.

### 4.5. Antimicrobial Susceptibility Testing (AST)

AST was carried out using VITEK2 (bioMérieux, Nürtingen, Germany). Testing was performed using software version 9.02 and AST-N389 card, according to the manufacturer’s instructions. The AST card used for the VITEK2 included an ESBL confirmation test. Second- and third-generation cephalosporins (ceftazidime, cefotaxime, and cefuroxime) were used alone or in combination with tazobactam. A reduction in growth in the presence of clavulanic acid was considered indicative of ESBL production.

MIC Breakpoints were set according to the European Committee on Antimicrobial Susceptibility Testing (EUCAST) breakpoint tables for interpretation of MICs and zone diameters (Version 11.0, 2021. http://www.eucast.org, accessed on 1 December 2021).

## Figures and Tables

**Table 1 antibiotics-11-00940-t001:** Genotypic characterization of sequenced ESBL-*E. coli* isolates; presence of a certain factor is based on the results from ABRicate [ABRicate v. 1.0.0 (https://github.com/tseemann/abricate, accessed on 1 April 2022), databases used: VFDB, ResFinder, PlasmidFinder, BacMet, ARG-ANNOT, and Ecoli_VF] using de novo-assembled sequences and is depicted in black. Detected genes are assigned to the following categories: ^1^ Plasmid replicon types, ^2^ aminoglycosides, ^3^ beta-lactam antibiotics, ^4^ phenicol antibiotics, ^5^ sulfonamides and trimethoprim, ^6^ tetracycline antibiotics, ^7^ macrolide, lincosamide, and streptogramin B.

Designation	1623	1624	1625	1626	1627	1628	1629	1630	1631	1632	1633	1634	1635	1636	1637	1638	1639	1640	1641	1642	1643	1644	1645	1646	1647	1648	1649
Origin	Calf	Calf	Calf	Calf	Calf	Calf	Heifer	Calf	Young stock	Calf	Calf	Calf	Calf	Calf	Calf	Calf	Calf	Calf	Calf	Calf	Calf	Calf	Young stock	Slurry	Young stock	Slurry	Slurry
Sequence Type	ST362	ST117	ST362	ST362	ST117	ST88	new *	ST362	ST967	ST362	ST362	ST362	ST362	ST362	ST362	ST362	ST362	ST362	ST362	ST362	ST117	ST362	ST967	ST117	ST967	new *	new **
Rearing Farm							+		+														+		+	+	+
IncB/O/K/Z ^1^																											
IncI ^1^																											
IncFIA ^1^																											
IncFIB ^1^																											
IncFII ^1^																											
IncH1																											
IncR ^1^																											
ColRNAI ^1^																											
Col156 ^1^																											
Col(MG828) ^1^																											
Col(pHAD28) ^1^																											
*aadA* ^2^																											
*aph3 ^2^*																											
*strA* ^2^																											
*strB* ^2^																											
*bla_CTX-M-1_* ^3^																											
*bla_CTX-M-15_* ^3^																											
*ampH* ^3^																											
AmpC1_*E. coli* ^3^																											
AmpC2_*E. coli* ^3^																											
*bla_TEM-105_* ^3^																											
*floR* ^4^																											
*catA* ^4^																											
*sul1* ^5^																											
*sul2* ^5^																											
*dfrA* ^5^																											
*tetA* ^6^																											
*tetB* ^6^																											
*tetR* ^6^																											
*tetY* ^6^																											
*mph(A)* ^7^																											

* highly identical with ST58; ** highly identical with ST10.

**Table 2 antibiotics-11-00940-t002:** Phenotypic resistance profiles of ESBL-*E. coli* isolates. R = resistant, S = sensitive.

Designation	Amoxicillin	Amoxicillin/ Clavulanic Acid	Ampicillin	Cefalexin	Cefotaxim	Ceftazidim	Ceftolozan/ Tazobactam	Cefepim	Colistin	Imipenem	Meropenem	Gentamicin	Tobramycin	Ciprofloxacin	Amikacin	Fosfomycin	Tigecycline	Trimethoprim/ Sulfamethoxazole	ESBL	MDR
1623	R	R	R	R	R	R	S	R	S	S	S	S	S	S	S	S	S	R	+	+
1624	R	R	R	R	R	R	S	R	S	S	S	S	S	R	S	S	S	S	+	+
1625	R	R	R	R	R	R	S	R	S	S	S	S	S	S	S	S	S	R	+	+
1626	R	R	R	R	R	R	S	R	S	S	S	S	S	S	S	S	S	R	+	+
1627	R	R	R	R	R	R	S	R	S	S	S	S	S	R	S	S	S	S	+	+
1628	R	R	R	R	R	R	S	R	S	S	S	S	S	S	S	S	S	R	+	+
1629	R	R	R	R	R	R	S	R	S	S	S	S	S	S	S	S	S	R	+	+
1630	R	R	R	R	R	R	S	R	S	S	S	S	S	S	S	S	S	R	+	+
1631	R	R	R	R	R	R	S	R	S	S	S	S	S	S	S	S	S	S	+	−
1632	R	R	R	R	R	R	S	R	S	S	S	S	S	S	S	S	S	R	+	+
1633	R	R	R	R	R	R	S	R	S	S	S	S	S	S	S	S	S	R	+	+
1634	R	R	R	R	R	R	S	R	S	S	S	S	S	S	S	S	S	R	+	+
1635	R	R	R	R	R	R	S	R	S	S	S	S	S	S	S	S	S	R	+	+
1636	R	R	R	R	R	R	S	R	S	S	S	S	S	S	S	S	S	R	+	+
1637	R	R	R	R	R	R	S	R	S	S	S	S	S	S	S	S	S	R	+	+
1638	R	R	R	R	R	R	S	R	S	S	S	S	S	S	S	S	S	R	+	+
1639	R	R	R	R	R	R	S	R	S	S	S	S	S	S	S	S	S	R	+	+
1640	R	R	R	R	R	R	S	R	S	S	S	S	S	S	S	S	S	R	+	+
1641	R	R	R	R	R	R	S	R	S	S	S	S	S	S	S	S	S	R	+	+
1642	R	R	R	R	R	R	S	R	S	S	S	S	S	S	S	S	S	R	+	+
1643	R	R	R	R	R	R	S	R	S	S	S	S	S	R	S	S	S	S	+	+
1644	R	R	R	R	R	R	S	R	S	S	S	S	S	S	S	S	S	R	+	+
1645	R	R	R	R	R	R	S	R	S	S	S	S	S	S	S	S	S	S	+	−
1646	R	R	R	R	R	R	S	R	S	S	S	S	S	R	S	S	S	S	+	+
1647	R	R	R	R	R	R	S	R	S	S	S	S	S	S	S	S	S	S	+	−
1648	R	R	R	R	R	R	S	R	S	S	S	S	S	S	S	S	S	R	+	+
1649	R	R	R	R	R	R	S	R	S	S	S	S	S	S	S	S	S	S	+	−

**Table 3 antibiotics-11-00940-t003:** Sample numbers for each animal group and slurry.

Animal Group	Number of Samples
Calves	20
Young stock *	20
Breeding heifers *	20
Dry cows	20
High-lactating cows	15
Late-lactating cows	15
Slurry **	10
Total	120

* Animals were kept on a separate rearing farm; ** Half of the samples originated from the main farm and half from the rearing farm.

## Data Availability

Data for this study were deposited in the European Nucleotide Archive (ENA) at EMBL-EBI under accession number PRJEB53375 (https://www.ebi.ac.uk/ena/browser/view/PRJEB53375, accessed on 20 June 2022).

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
