# Peer review of "Distribution of ESBL/AmpC-Escherichia coli on a Dairy Farm"

_antibiotics, 2022, doi:10.3390/antibiotics11070940_

Round 1
Reviewer 1 Report
The authors presented a manuscript intitled "Distribution of ESBL/AmpC-Escherichia coli on a dairy farm" were are reporting 27 E. coli isolates with phenotypic cefotaxime resistance. Methods are well performed and the results are of importance to the field.
My suggestions are as follows:
1. E. coli must be written italicized in line 45, please review the entire document
2. Sentences between lines 49-76 sounds like discussion, I suggest modify redaction or shift those sentences to discussion.
3. What is the importance of the different plasmids you've found? This is a significant finding and it feels like something weak, I suggest to highlight it.
4. In line 205, redaction must be changed to "Since no ESBL/AmpC-E.coli strains were isolated from animal adults..."
5. May be interesting to determine the effect of the combination of the CTX genes detected in some samples to resistance phenotype. I suggest to implement Minimum Inhibitory Concentration tests.
Author Response
The authors presented a manuscript intitled "Distribution of ESBL/AmpC-Escherichia coli on a dairy farm" were are reporting 27 E. coli isolates with phenotypic cefotaxime resistance. Methods are well performed and the results are of importance to the field.
Thank you.
My suggestions are as follows:
- E. coli must be written italicized in line 45, please review the entire document
Apologize. Corrected and checked throughout the manuscript.
- Sentences between lines 49-76 sounds like discussion, I suggest modify redaction or shift those sentences to discussion.
Thank you. We would prefer to leave this passage in the introductory section, as the cited studies were the reason for us to conduct the present study. The cited ones also formed the basis for determining the sample size.
- What is the importance of the different plasmids you've found? This is a significant finding and it feels like something weak, I suggest to highlight it.
Thank you. We included this in the Discussion section (lines 263-267).
- In line 205, redaction must be changed to "Since no ESBL/AmpC-E.colistrains were isolated from animal adults..."
Apologize. Corrected.
- May be interesting to determine the effect of the combination of the CTX genes detected in some samples to resistance phenotype. I suggest to implement Minimum Inhibitory Concentration tests.
Good point. However, there were no differences between CTX-M-1- and CTX-M-15 producing isolates with respect to MIC values.
Reviewer 2 Report
Thank you for the opportunity to review this paper. The authors of this manuscript examined the prevalence of ESBL/AmpC-E. coli among fecal samples collected from cattle on a farm, and to develop targeted interventions. The analysis is simple, but the authors spent some time finding the reasons behind the resistance in the discussion, which could be useful. I recommend publish the paper after minor revision.
Feedback by section:
Introduction:
- Please add one or two sentences about the rationale of studying resistance in dairy farming after introducing the importance of ESBL-producing E. coli.
- “In 2016, 58.6% human-derived E. coli isolates already showed resistance to at least one antibiotic agent” Please include more context of this number. Where’re the samples collected, who’re the study population.
- “In total, 39 of 45 participating farms ESBL/AmpC producing E. coli were present” Italicize E., and rewrite the sentence as “ESBL/AmpC producing E. coli were present were present in 39 of 45 participating farms”
Results:
- The authors should at least do a chi-square/CMH test to test the differences of prevalence of ESBL/AmpC E. coli between cattle of different age groups, as it is one of the highlighted results.
- Add acronym MDR in results before using it in Table 2.
Discussion:
- Same as my previous comment, the authors need a statistical test to claim “significantly higher proportion of resistant pathogens than in all other age groups sampled”
Author Response
Thank you for the opportunity to review this paper. The authors of this manuscript examined the prevalence of ESBL/AmpC-E. coli among fecal samples collected from cattle on a farm, and to develop targeted interventions. The analysis is simple, but the authors spent some time finding the reasons behind the resistance in the discussion, which could be useful. I recommend publish the paper after minor revision.
Thank you.
Feedback by section:
Introduction:
Please add one or two sentences about the rationale of studying resistance in dairy farming after introducing the importance of ESBL-producing E. coli.
Done (lines 45-46)
In 2016, 58.6% human-derived E. coli isolates already showed resistance to at least one antibiotic agent” Please include more context of this number. Where’re the samples collected, who’re the study population.
We added information on the collection period, the study population and a reference. (lines 42-44)
In total, 39 of 45 participating farms ESBL/AmpC producing E. coli were present” Italicize E., and rewrite the sentence as “ESBL/AmpC producing E. coli were present were present in 39 of 45 participating farms”
Done and checked throughout the manuscript.
Results:
The authors should at least do a chi-square/CMH test to test the differences of prevalence of ESBL/AmpC E. coli between cattle of different age groups, as it is one of the highlighted results.
Yes, thank you. We added a statistical analysis (lines 103-104, 106-108, 471-472).
Add acronym MDR in results before using it in Table 2.
Thank you, we introduced this abbreviation (line 204)
Discussion:
Same as my previous comment, the authors need a statistical test to claim “significantly higher proportion of resistant pathogens than in all other age groups sampled”
Thank you, we added a statistical analysis (lines 103-104, 106-108, 471-472).
Reviewer 3 Report
This study aimed to determine the prevalence of ESBL/AmpC-producing Escherichia coli and to investigate their on-farm distribution on an exemplary dairy farm. The intensive cleaning with disinfectants appeared crucial for reducing the spread of ESBL/AmpC-producing E. coli in calves while the frequent use of antibiotics in animal farming could contribute to the increased AMR potentials of these bacteria. Despite these important findings, the authors must address some areas of concern.
Areas of concern:
General
The results and methods should be reported in the past tense.
There is a need for English language editing.
Abstract
This section lacks the background and information on the sample design as well as the microbiological isolation and detection techniques.
Introduction
Lines 34-36: it is important to mention other factors of increased antimicrobial resistance apart from the use of antibiotics in animal husbandry
Line 45: The name of the bacteria needs to be in italic
Lines 70-76: This section should come before lines 67-69.
Concerning lines 67-69, the authors should indicate that this was to refer specifically to German dairy farms.
Results
Lines 102-104: The authors are repeating the methodology here. This section should be moved to the materials and methods section.
Lines 115-117: This statement seems to contract the information contained in line 144.
Line 159-160: This sentence is not complete. Please verify this.
Lines 156-167: Whole genome analysis of AMRGs should come after Phenotypic AST to verify congruence
Table 1: This Table should be improved. For example, the grey or dark color in the boxes should be changed to blue or green color for better visibility. Moreover, except for the title, the explanations contained in lines 171-177 should come at the end of the Table and not before.
Discussion
Lines 107-110: the potential risk factors for the transmission of ESBL/AmpC-E.coli reported in the result section were not discussed in the discussion section
Lines 215-216 and 226: The dominance of ST362 in calves could not be attributed to increased resistance to biocides since detergent and disinfectant were not used in this study
Lines 267-268: I think, the authors meant ‘’animals’’ and not ‘’ animals humans’’
Lines 268-271: This statement is unclear.
Lines 281-283: this statement is equally unclear
Recommendation
Lines 330-353+357-362+364-367: the recommendation section is poorly written. The selected section of the recommendation should be integrated into the discussion section.
Conclusion
This study does not have a conclusion
Materials and methods
In general, why did you collect 120 samples in total? Is there any rationale for it?
What is the rationale for your sample size distribution in Table 3?
What was your study duration? Which period?
Did you use dry or wet swabs?
What was the concentration of the purified DNA used for library preparation?
Lines 417-422: specify the algorithms used for the different bio-informatics analyses
Antimicrobial susceptibility testing should be performed before Whole genome analysis of AMRGs
No statistical analysis is mentioned in this study despite the fact you collected samples from different animal groups, which could have constituted an important source of variation.
Author Response
This study aimed to determine the prevalence of ESBL/AmpC-producing Escherichia coli and to investigate their on-farm distribution on an exemplary dairy farm. The intensive cleaning with disinfectants appeared crucial for reducing the spread of ESBL/AmpC-producing E. coli in calves while the frequent use of antibiotics in animal farming could contribute to the increased AMR potentials of these bacteria. Despite these important findings, the authors must address some areas of concern.
Thank you.
Areas of concern:
General
The results and methods should be reported in the past tense.
Done.
There is a need for English language editing.
Thank you. We have carefully revised the English and corrected where necessary. We have completely revised some passages.
Abstract
This section lacks the background and information on the sample design as well as the microbiological isolation and detection techniques.
Thank you. Information added (lines 11-12).
Introduction
Lines 34-36: it is important to mention other factors of increased antimicrobial resistance apart from the use of antibiotics in animal husbandry
Thank you. We added the following lines: Other factors that influence the occurrence of antibiotic resistance include farm management, water treatment, manure handling, and wildlife control (lines 36-38).
Line 45: The name of the bacteria needs to be in italic
Changed, thank you. We check this throughout the manuscript.
Lines 70-76: This section should come before lines 67-69.
Concerning lines 67-69, the authors should indicate that this was to refer specifically to German dairy farms.
Changed and clarified (lines 72-73). Thank you.
Results
Lines 102-104: The authors are repeating the methodology here. This section should be moved to the materials and methods section.
Yes, thank you. We removed these lines.
Lines 115-117: This statement seems to contract the information contained in line 144.
Thank you. We clarified this.
Line 159-160: This sentence is not complete. Please verify this.
Thank you. Corrected.
Lines 156-167: Whole genome analysis of AMRGs should come after Phenotypic AST to verify congruence
Thank you. We can understand this point, nevertheless we would like to leave the sequence in its original form. We would leave the decision to the editor.
Table 1: This Table should be improved. For example, the grey or dark color in the boxes should be changed to blue or green color for better visibility. Moreover, except for the title, the explanations contained in lines 171-177 should come at the end of the Table and not before.
Thank you. For ease of printing, we would prefer to leave the table in black and white. The position of the legend is given in the template of antibiotics.
Discussion
Lines 107-110: the potential risk factors for the transmission of ESBL/AmpC-E.coli reported in the result section were not discussed in the discussion section
Good point, thank you. We have added this to the discussion (lines 233-234, 237-243, 246-256).
Lines 215-216 and 226: The dominance of ST362 in calves could not be attributed to increased resistance to biocides since detergent and disinfectant were not used in this study
Further phenotypic studies on the biofilm formation capacity and biocide resistance of the ST362 isolates in this study are needed for clarification (lines 292-294).
Lines 267-268: I think, the authors meant ‘’animals’’ and not ‘’ animals humans’’
Thank you, corrected.
Lines 268-271: This statement is unclear.
Rephrased
Lines 281-283: this statement is equally unclear
Rephrased
Recommendation
Lines 330-353+357-362+364-367: the recommendation section is poorly written. The selected section of the recommendation should be integrated into the discussion section.
Thank you, we removed this section. Some parts of this were rephrased and integrated in the first part of the discussion (lines 225-256).
Conclusion
This study does not have a conclusion
Yes, thank you. We integrated the following conclusion: On our study farm, we found a very high prevalence of ESBL-producing E. coli in the calf age group. Using WGS, we were able to demonstrate that these ESBL E. coli most likely did not originate from the dams but from the calves' environment. Due to weaknesses in calf feeding hygiene, we recommend a targeted improvement of hygiene in this area to reduce the load of these bacteria. (lines 432-437)
Materials and methods
In general, why did you collect 120 samples in total? Is there any rationale for it?
The sample size was calculated using the https://epitools.ausvet.com.au/oneproportion sample size calculation tool, and based on the number of appropriately aged animals present on the day of sampling. According to our previous study, we chose to use an expected ESBL-E. coli prevalence of 60 % for the age group of calves, and 20 % for the other animals, a confidence interval of 90 %, and a desired precision of 10 %. In cows, we focused on lactating animals (high and late lactating) during sampling.
What is the rationale for your sample size distribution in Table 3?
See above
What was your study duration? Which period?
Thank you. All samples were collected on 1st of August 2021. We added this in the Methods and Material section (line 448)
Did you use dry or wet swabs?
We used swabs with amies transgport medium. We added this in the Methods and Material section (line 450)
What was the concentration of the purified DNA used for library preparation?
The concentration of the purified DNA was at least 10ng/µL, as recommended by sequencing center.
Lines 417-422: specify the algorithms used for the different bio-informatics analyses
Details on the sequence analysis are given in our previous publications (10.3390/antibiotics10050568 and 10.3390/antibiotics11020123).
Antimicrobial susceptibility testing should be performed before Whole genome analysis of AMRGs
Thank you. We can understand this point, nevertheless we would like to leave the sequence in its original form. We would leave the decision to the editor.
No statistical analysis is mentioned in this study despite the fact you collected samples from different animal groups, which could have constituted an important source of variation.
Thank you, we added a statistical analysis (lines 103-104, 106-108, 471-472).